# Favorable Effect of Pemafibrate on Insulin Resistance and β-Cell Function in Subjects with Type 2 Diabetes and Hypertriglyceridemia: A Subanalysis of the PARM-T2D Study

**DOI:** 10.3390/pharmaceutics15071838

**Published:** 2023-06-27

**Authors:** Hiroshi Nomoto, Kenichi Kito, Hiroshi Iesaka, Yuki Oe, Shinichiro Kawata, Kazuhisa Tsuchida, Shingo Yanagiya, Aika Miya, Hiraku Kameda, Kyu Yong Cho, Ichiro Sakuma, Naoki Manda, Akinobu Nakamura, Tatsuya Atsumi

**Affiliations:** 1Department of Rheumatology, Endocrinology and Nephrology, Faculty of Medicine and Graduate School of Medicine, Hokkaido University, Sapporo 060-8638, Hokkaido, Japan; 2Caress Sapporo Hokko Memorial Clinic, Sapporo 065-0027, Hokkaido, Japan; 3Manda Memorial Hospital, Sapporo 060-0062, Hokkaido, Japan

**Keywords:** insulin resistance, pemafibrate, type 2 diabetes

## Abstract

Pemafibrate, a novel selective peroxisome proliferator-activated receptor modulator, has beneficial effects on lipid metabolism. However, its effects on glucose metabolism in individuals with type 2 diabetes (T2DM) remain to be fully clarified. This was a subanalysis of the PARM-T2D study, a multicenter prospective observational study on the use of pemafibrate versus conventional therapy for 52 weeks in subjects with T2DM complicated with hypertriglyceridemia. The subanalysis included participants who did not change their treatment for diabetes and did not receive insulin or insulin secretagogues during the study period. Changes in glucose metabolism markers, including homeostatic model assessment (HOMA2) scores and disposition index, were assessed. A total of 279 participants (141 in the pemafibrate group; 138 in the control group) met the criteria for the subanalysis. There were no significant changes in HbA1c during the 52-week study period in both groups. However, the pemafibrate group showed significant improvements versus the control group for insulin resistance assessed by HOMA2-R (−0.15 versus 0.08; estimated treatment difference −0.23 (95% confidence interval −0.44, −0.02); *p* = 0.03) and maintenance of β-cell function assessed by disposition index (0.015 versus −0.023; estimated treatment difference 0.037 (95% confidence interval 0.005, 0.069); *p* = 0.02). Correlation analyses showed that improvements in HOMA2-R and disposition index were significantly associated with improvements in lipid abnormalities and γ-glutamyl transpeptidase. In conclusion, pemafibrate reduced insulin resistance and maintained β-cell function in subjects with T2DM and hypertriglyceridemia, presumably by improving lipid profiles and lipid-related hepatocyte stress.

## 1. Introduction

Type 2 diabetes (T2DM), one of the most important non-communicable diseases, is caused by complex and multiple triggers including genetic and environmental factors. Subjects with T2DM are at a markedly higher risk of cardiovascular events compared with those without diabetes [1], and maintaining fair glycemic control can reduce such risks [2]. In addition, insufficient glycemic control might be related to mortality in subjects with diabetes [3]. Therefore, the maintenance of physiological blood glucose levels is important to prevent cardiovascular events and death related to diabetes/hyperglycemia.

Insulin resistance and reduced insulin secretory capacity are core pathophysiological features of T2DM [4,5]. Insulin resistance occurs before the onset of diabetes. Although β-cells initially secrete additional insulin to compensate for the relative insulin deficiency, the β-cell function deteriorates as T2DM progresses [6,7,8]. Because high insulin demand arising from insulin resistance may be one of the underlying reasons for β-cell failure, treatment strategies that can increase insulin sensitivity are desired.

Dyslipidemia, a frequent complication of T2DM and a risk factor for mortality in T2DM subjects [3,9], has a close relationship with insulin resistance [10,11]. Among several therapeutic strategies for dyslipidemia, fibrates effectively reduce serum triglyceride (TG) and increase serum high-density lipoprotein cholesterol (HDL-C) by activating peroxisome proliferator-activated receptors (PPARs) [12]. Recently, pemafibrate, a selective PPARα modulator, was identified as a potent and highly selective agonist for human PPARα [13]. A phase 3 clinical trial demonstrated the superiority of pemafibrate over fenofibrate for lowering serum TG in patients with dyslipidemia [14]. Pemafibrate also reduced the homeostatic model assessment (HOMA)-insulin resistance score compared with a placebo in subjects with T2DM comorbid with hypertriglyceridemia who did not receive insulin sensitizers [15]. However, its long-term efficacy has not been verified in real-world clinical settings, wherein patients frequently receive treatment with several anti-diabetic agents, including biguanides and sodium–glucose cotransporter 2 (SGLT2) inhibitors. Considering such anti-diabetic agents that are essential for managing diabetes and cardiovascular outcomes have been used widely [16], clinical trials targeting subjects with T2DM treated with these insulin sensitizers are warranted. We previously conducted a prospective observational study evaluating the efficacy and safety of pemafibrate compared with conventional treatments for the improvement in lipid profile and other metabolic parameters in patients with T2DM complicated with hypertriglyceridemia in real-world clinical practice (PARM-T2D study). That study revealed the strong efficacy of pemafibrate on improving lipid profiles, liver and renal functions may lead to improved glucose metabolism [17]. Here, we aimed to clarify the effects of pemafibrate on glucose metabolism based on our previous PARM-T2D study assessing the effect of pemafibrate on lipid profiles in subjects with T2DM.

## 2. Materials and Methods

### 2.1. Study Design and Participants

This was a secondary analysis of our previous multicenter prospective observational PARM-T2D study comparing the efficacy and safety of pemafibrate with conventional therapies [17]. Briefly, 685 patients with T2DM and hypertriglyceridemia aged ≥ 20 years who were fibrate-naive or taking conventional fibrates were enrolled in the original study. The major exclusion criteria were as follows: allergy to pemafibrate, pregnant women, serious liver and renal dysfunction, and other reasons including incompatibility with the study. In the pemafibrate group, pemafibrate 0.2–0.4 mg/day was initiated in fibrate-naive patients or switched in patients taking conventional fibrates. In the control group, patients continued taking their fibrates or were not receiving any medications for hypertriglyceridemia. Fasting blood/urine samples and physical assessments were evaluated at baseline and weeks 12, 24, and 52. As a marker for endogenous insulin secretion, fasting serum C-peptides were measured at baseline and weeks 24 and 52. The changes in glycated hemoglobin (HbA1c), HOMA2-R, HOMA2-β, and disposition index during the study period were compared between the pemafibrate group and the control group for evaluation of glucose metabolism. The HOMA2 scores were generated using HOMA2 Calculator Version 2.2.3 (available from www.dtu.ox.ac.uk/homacalculator/, accessed on 24 August 2022). The disposition index, reflecting the ability of β-cells to compensate for insulin resistance, was calculated by multiplying HOMA2-S by HOMA2-β as described elsewhere [18].

The PARM-T2D study was registered with the University Hospital Medical Information Network (UMIN) Center Clinical Trials Registry (UMIN000037385). The protocol was approved by the Institutional Review Board of Hokkaido University Hospital Clinical Research and Medical Innovation Center (018-0440), and the study was performed in accordance with the principles of the Declaration of Helsinki and its amendments.

The original study was conducted at nine specialized centers for the treatment of diabetes located in Hokkaido, Japan (PARM-T2D study cohort). The participants were treated at each medical center throughout the study period. Patients who were treated with insulin injection therapy, were taking sulfonylureas or glinides, or changed their treatment regimens for diabetes were excluded from the secondary analysis to avoid confounding effects on the indices for glucose metabolism, as recommended in a previous report [19]. Patients with extremely low or high C-peptide levels (≤0.6 or >10.5 ng/mL, respectively) were also excluded because the HOMA scores could not be calculated in these cases.

### 2.2. Statistical Analysis

Normally distributed data were expressed as mean ± SD, and non-normally distributed data were expressed as median (25% percentile, 75% percentile) for continuous variables or number (proportion) for categorical variables. Differences between the two groups were compared using an unpaired *t*-test for continuous variables and a chi-square test or Fisher’s exact test for categorical variables. Data within the groups were compared by a paired *t*-test or the Wilcoxon signed-rank test. Because the efficacy of pemafibrate on metabolic parameters can be affected by prior use of conventional fibrates, we also conducted an analysis of covariance (ANCOVA) to adjust for these confounders. To clarify the clinical features of the patients who received the merits of pemafibrate on glucose metabolism, we divided participants into two groups (improved group and deteriorated group) based on the changes of each index reflecting glucose metabolism for 52 weeks. In addition, correlations between changes in indices for glucose metabolism associated with pemafibrate and changes in metabolic parameters in the 52-week study period between were also evaluated by Spearman’s rank correlation analysis. Multivariate analyses were carried out using multiple linear regression to identify factors independently associated with the outcomes. Data were analyzed using GraphPad Prism 8.4.2 (GraphPad Software Inc., San Diego, CA, USA) or JMP Pro 16.0.0 (SAS Inc., Cary, NC, USA). *p* < 0.05 indicated statistical significance.

## 3. Results

A total of 685 patients were enrolled, of whom 650 met the inclusion criteria for the PARM-T2D study. From this original cohort, 35 participants who did not meet the inclusion criteria and 268 patients who were not suitable for this subanalysis, mainly because of changes in medications for comorbidities that can affect glucose metabolism, lack of relevant data, and/or an interruption in hospital visits, were excluded. Thereafter, 103 patients who were treated with insulin, sulfonylureas, and/or glinides were excluded based on the recommendation for use of HOMA-indices [19]. As a result, 279 participants (141 in the pemafibrate group; 138 in the control group) who were not treated with insulin or insulin secretagogues and had the full set of relevant data available, including serum C-peptide, met the criteria for the subanalysis (Figure 1). There were no significant differences in the baseline characteristics between the two groups, including glycemic control, liver and kidney function, and serum lipid profiles (Table 1). A breakdown of concomitant dedication for diabetes showed that no participants were treated with insulin or insulin secretagogues in this subanalysis. Two-thirds of the subjects were treated with biguanides, and there were no significant differences in the proportion of baseline treatment with SGLT2 inhibitors and GLP-1 receptor agonists, which might potently affect insulin resistance and β-cell function (Table 1). Notably, one-third of the patients were treated with conventional fibrates in both groups. Fibrates were switched to pemafibrate in the pemafibrate group, whereas these conventional fibrates were continued in the control group, as shown in the original study [17].

After 52 weeks of treatment, HOMA2-R, which mainly reflects insulin resistance in the liver, showed a significant improvement in the pemafibrate group only (2.11 to 1.92 (pemafibrate) versus 1.99 to 1.94 (control)) (*p* = 0.017). Regarding β-cell function, there were no significant changes in HOMA2-β in the two groups (75.0 to 69.4 (pemafibrate) versus 75.9 to 71.3 (control)), but the pemafibrate group had a slightly increased disposition index (*p* = 0.088), which was significantly different compared with the control group (+0.02 (pemafibrate) versus −0.01 (control)) (Table 2). These changes were not significant at 24 weeks. Considering that at baseline each value was slightly different, and glycemic control and the use of conventional fibrate can affect these indices, we additionally validated these changes using ANCOVA. These differences between the groups in HOMA2-R and the disposition index were verified by ANCOVA adjusted for each baseline parameter and fibrate use (Figure 2). Focusing on insulin resistance, we explored the patient characteristics associated with the improvement in HOMA2-R after pemafibrate treatment by categorizing the pemafibrate group into two subgroups: improved group (ΔHOMA2-R < 0, *n* = 83) and deteriorated group (ΔHOMA2-R ≥ 0, *n* = 58). As shown in Appendix A, the improved group had higher baseline fasting plasma glucose (FPG) (140.1 ± 32.1 mg/dL versus 128.3 ± 26.3 mg/dL), γ-glutamyl transpeptidase (γ-GTP) (45 (31, 84) IU/L versus 34 (22, 56) IU/L), and HOMA2-R (2.30 (1.74, 3.14) versus 1.75 (1.32, 2.37)) compared with the deteriorated group (*p* = 0.023, *p* = 0.007, and *p* < 0.001, respectively). Similar analysis focusing on β-cell function assessed by the disposition index revealed that patients who improved the disposition index showed significantly higher FPG (142.6 ± 34.8 mg/dL versus 126.8 ± 19.8 mg/dL), HbA1c (7.08 ± 0.88% versus 6.72 ± 0.59%), and HOMA2-R (2.30 (1.66, 3.09) versus 1.97 (1.39, 2.59)), whereas the baseline disposition index and rate of fibrate pretreatment were significantly lower in the disposition index improved group (Appendix A). Importantly, the extent of changes in HOMA2-R and the disposition index were not affected by the regimens of concomitant anti-diabetic agents (Appendix A).

Similar to the findings in the original cohort, pemafibrate significantly improved the lipid profiles and liver dysfunction: an increase in HDL-C by +2.0 (95% confidence interval (CI): 0.3 to 3.8) mg/dL, and decreases in TG, aspartate aminotransferase, alanine aminotransferase, and γ-GTP by −46 (−60 to −30) mg/dL, −3.5 (−5.4 to −1.6) IU/L, −6 (−9 to −4) IU/L, and −10 (−14 to −5) IU/mL, respectively (Appendix A). Glycemic control parameters, such as FPG and HbA1c, remained unchanged during the study period in both the pemafibrate group and the control group (Table 2). Among the various metabolic parameters, changes in TG, HDL-C, and γ-GTP were positively correlated with change in HOMA2-R in the pemafibrate group, while negative correlations were found between a change in the disposition index and changes in body mass index (BMI), TG, and γ-GTP (Table 3). Multiple regression analyses showed that an increase in HDL-C was significantly associated with improvement in HOMA2-R (*p* = 0.002) and changes in TG and BMI were associated with improvement in the disposition index (*p* = 0.011 and *p* = 0.022) in the pemafibrate group (Appendix A).

## 4. Discussion

In this secondary analysis of the prospective observational PARM-T2D trial on the use of pemafibrate in adults with hyperglycemia complicated with T2DM, administration of pemafibrate significantly improved insulin resistance as well as β-cell function assessed by HOMA2-R and the disposition index. A preferable effect of pemafibrate on insulin resistance was found in a previous pooled meta-analysis involving short-term phase 2 and 3 trials investigating the efficacy of pemafibrate compared with a placebo in patients affected by dyslipidemia with or without T2DM [20]. The strengths of the present subanalysis were: (1) targeting the T2DM population only, (2) including participants who were treated with several insulin sensitizers, reflecting real-world clinical settings, (3) excluding subjects who changed their anti-diabetic medications during the study period, and (4) having a relatively long-term study design (52 weeks). In addition, clinical parameters related to improvement of glucose metabolism indices, which were not examined in previous phase 2 or 3 trials, were explored.

The efficacy of conventional fibrates for glucose metabolism has been controversial, particularly in people with T2DM [21,22,23,24]. A recent meta-analysis involving 22 randomized placebo-controlled trials found that fibrate use significantly decreased both FPG and insulin resistance assessed by HOMA-R, but not HbA1c [25]. Fibrates were suggested to exert this efficacy through effects on lipid metabolism and anti-inflammatory effects [26]. Pemafibrate has a potent ability to improve lipid metabolism, even compared with fibrates [17,27], and has a similar anti-inflammatory mechanism related to fibrates [28]. Another distinct feature of pemafibrate is its action on hepatic metabolism, which can be explained by increased levels of fibroblast growth factor 21 (FGF21). FGF21, a hormone primarily expressed by the liver and adipose tissue, is closely related to hepatic metabolic pathways [29]. Phase 3 clinical trials verified that serum FGF21 was elevated in pemafibrate-treated groups [15], possibly leading to improved liver function [17,20] as well as improvements in inflammation and steatosis of the liver [30,31]. In the present subanalysis, the reduction in HOMA2-R was significantly correlated with improvement of lipid metabolism and reduction of γ-GTP (Table 3). In addition, patients who had liver dysfunction and higher insulin resistance showed improvement of HOMA2-R after pemafibrate treatment (Appendix A). Considering the close relationship between liver dysfunction, including nonalcoholic fatty liver disease, and insulin resistance [32], the potent action of pemafibrate on improving HOMA2-R appears reasonable.

As well as ameliorating insulin resistance, pemafibrate resulted in preserved β-cell function assessed by the disposition index compared with the control group in the subanalysis. The control group showed a slight decrease in the disposition index during the 52-week study period, compatible with the decline in β-cell function over time [7]. To date, there has been no direct evidence for the effects of pemafibrate on pancreatic β-cells. Because there were no improvements in glycemic control parameters and no changes in the treatment regimens for T2DM, the improvements in insulin resistance and lipotoxicity may partially contribute to a reduced burden on β-cells, leading to preservation of β-cell function [33]. Indeed, reductions in TG and BMI, which can both lead to a β-cell burden, were significantly correlated with an improved disposition index in the subanalysis. Interestingly, patients having higher insulin resistance and a lower disposition index benefited from pemafibrate treatment on an improved disposition index, and such efficacy was obvious in fibrate-naive patients (Appendix A). Therefore, fibrate-naive subjects with T2DM and relatively high FPG and insulin resistance might derive a benefit related to glucose metabolism when administrated pemafibrate. In this subanalysis, the efficacy of pemafibrate on HOMA2-R and the disposition index was confirmed at 52 weeks but not at 24 weeks. However, an improvement in lipid metabolism and liver dysfunction was observed even at 12 and 24 weeks, as shown in the original PARM-T2D study [17]. Although the precise mechanism is not clear, considering that HOMA2-R reflects insulin resistance in the liver and the improvement of liver stiffness induced by pemafibrate was only observed after 48 weeks of treatment compared with a placebo in a phase 2 trial [34], it might take time to have an effect on insulin resistance and β-cell protection. To clarify the interactions between the biological parameters, histological changes, and insulin resistance in the liver, further clinical investigation, especially in non-alcoholic liver steatosis (NASH) and/or nonalcoholic fatty liver disease (NAFLD) cases, is needed. The reason for the discrepancy between the HOMA2-β and disposition index results can be explained by differences in the formulas used for the calculations. HOMA2-β consists of FPG and fasting C-peptide. Because fasting C-peptide can be regulated by the presence of insulin resistance, insulin secretory ability assessment by the disposition index, which takes account of insulin resistance, would be the more suitable method for our analysis.

The limitations of the original study were described previously [17]. The open-label observational study design can yield selection bias, and only Japanese patients were included in the study. In addition, the dose of pemafibrate was decided by the physicians in charge. The present subanalysis had additional limitations, mainly arising from the study design for the secondary analysis. Because we selected the patients from the original cohort using strict criteria to avoid confounders, the sample size was smaller than that of the original cohort. However, there were no significant differences in the patient background characteristics between the two groups, and the main effects of pemafibrate, such as improvements in lipid profiles and liver dysfunction, were similar to those in the original study. In addition, the subanalysis included patients treated with fibrates. To manage this issue, we conducted an ANCOVA adjusted for fibrate use and confirmed that the results were robust. Another potential limitation was the concomitant use of anti-diabetic medications. Although subjects who changed their treatment regiments during the study period were excluded from this subanalysis, we do not have information regarding the period of each anti-diabetic treatment before enrollment in our trial. However, the extent of changes related to baseline antidiabetic medications in HOMA2-R and the disposition index were similar. A further randomized controlled trial of pemafibrate in the T2DM population focusing on glucose metabolism markers is required in the future. In addition, an analysis with a focus on subjects with NASH/NAFLD complicated with T2DM is desired.

In conclusion, pemafibrate can ameliorate insulin resistance and retain β-cell function in subjects with T2DM and hypertriglyceridemia, which may be correlated with improved lipid profiles and lipid-related hepatocyte stress.

## Figures and Tables

**Figure 1 pharmaceutics-15-01838-f001:**
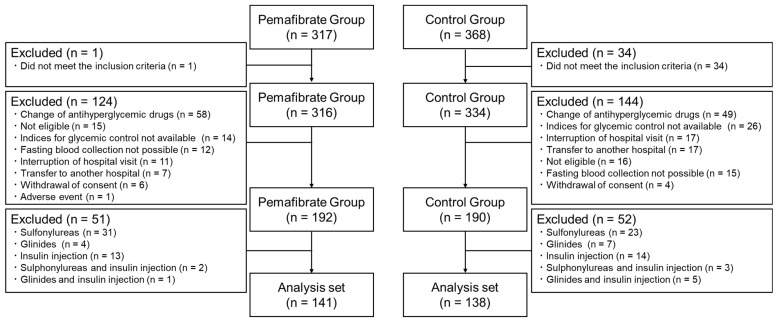
Flow diagram for the subanalysis. From the original cohort, subjects who did not meet the inclusion criteria, changed antihyperglycemic drugs during the study period, discontinued the study, or were treated with insulin and/or insulin secretagogues were excluded from the subanalysis.

**Figure 2 pharmaceutics-15-01838-f002:**
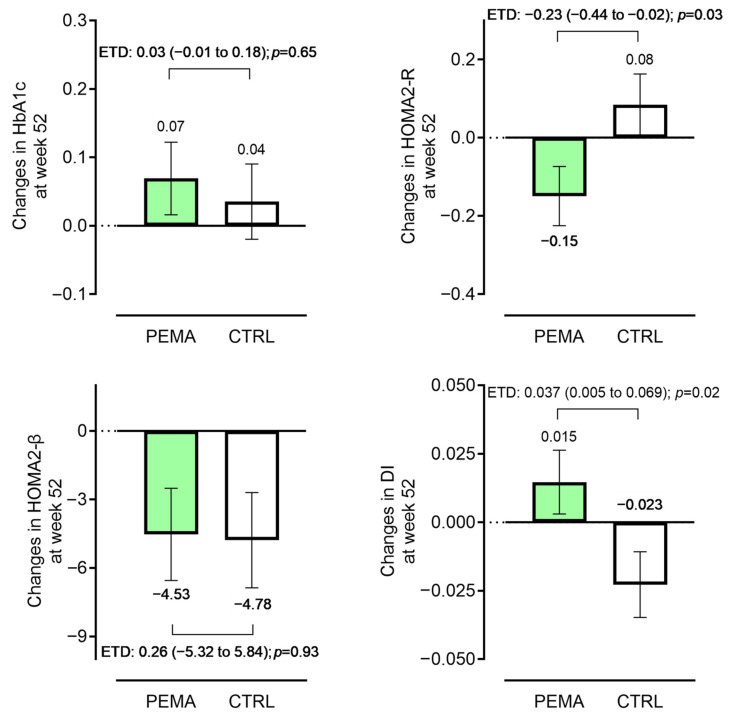
Changes in key endpoints from baseline. Changes in glycated hemoglobin (HbA1c), homeostatic model assessment (HOMA) 2-R, HOMA2-β, and disposition index (DI) during the 52-week study period (pemafibrate group versus control group). Data were adjusted for ANCOVA (covariates: baseline HbA1c and fibrate use for HbA1c; each baseline value, HbA1c, and fibrate use for HOMA2-R, HOMA2-β, and DI, respectively). Bars represent adjusted mean ± standard error. Differences between the two groups are shown as the estimated treatment difference (ETD) (95% confidence interval). CTRL, control; PEMA, pemafibrate.

**Table 1 pharmaceutics-15-01838-t001:** Demographic and clinical characteristics of the participants at baseline.

Variables	Pemafibrate (*n* = 141)	Control (*n* = 138)	*p*-Value
Age (years)	60.1 ± 12.3	60.9 ± 11.6	0.573
Female sex (*n*, %)	45 (31.9)	54 (39.1)	0.214
Body mass index (kg/m^2^)	27.3 ± 4.1	27.4 ± 4.8	0.932
HbA1c (%)	6.92 ± 0.78	6.83 ± 0.59	0.254
FPG (mg/dL)	135.2 ± 30.3	132.5 ± 26.3	0.423
C-peptide (mg/dL)	2.87 ± 1.32	2.84 ± 1.53	0.840
T-Cho (mg/dL)	185.0 ± 31.7	186.1 ± 32.8	0.732
Triglyceride (mg/dL)	171 (133, 235)	168 (126, 226)	0.708
HDL-C (mg/dL)	52.4 ± 11.9	52.0 ± 13.7	0.780
AST (IU/L)	31.5 ± 15.7	28.2 ± 14.5	0.069
ALT (IU/L)	30 (20, 45)	28 (17, 39)	0.102
γ-GTP (IU/L)	39 (24, 72)	41 (23, 56)	0.349
eGFR (mL/min/1.73 m^2^)	69.6 ± 18.3	69.7 ± 22.1	0.878
Fibrates (*n*, %)	52 (36.9)	43 (31.2)	0.377
α-glucosidase inhibitors (*n*, %)	2 (1.4)	2 (1.5)	1.000
Biguanides (*n*, %)	90 (63.8)	87 (63.0)	0.902
DPP-4 inhibitors (*n*, %)	66 (46.8)	65 (47.1)	1.000
GLP-1 receptor agonists (*n*, %)	7 (5.0)	10 (7.2)	0.463
SGLT2 inhibitors (*n*, %)	66 (46.8)	49 (35.5)	0.068
Thiazolidines (*n*, %)	7 (5.0)	3 (2.2)	0.335
Sulfonylureas (*n*, %)	0	0	NA
Glinides (*n*, %)	0	0	NA
Insulin injections (*n*, %)	0	0	NA

Data are shown as the mean ± SD, median (25% percentile, 75% percentile), or number (%). *p*-values for the pemafibrate group versus control group were obtained using the Student’s *t*-test, the Mann–Whitney U-test, or Fisher’s exact test. HbA1c, glycated hemoglobin; FPG, fasting plasma glucose; T-Cho, total cholesterol; HDL-C, high-density lipoprotein cholesterol; AST, aspartate aminotransferase; ALT, alanine aminotransferase; γ-GTP, γ-glutamyl transpeptidase; eGFR, estimated glomerular filtration rate, DPP-4, dipeptidyl peptidase-4; GLP-1, glucagon-like peptide-1; SGLT2, sodium glucose cotransporter 2; NA, not assessed.

**Table 2 pharmaceutics-15-01838-t002:** Changes in parameters for glycemic control, insulin resistance, and insulin secretion by week.

		Week 0	Week 24	Week 52	Mean Change at Week 52	*p*-Value between Groups at Week 52
HbA1c (%)	PEMA (*n* = 141)	6.92 ± 0.78	7.03 ± 0.93	6.99 ± 0.81	0.06(−0.04 to 0.16)	0.940
	CTRL (*n* = 138)	6.83 ± 0.59	6.94 ± 0.76	6.89 ± 0.80	0.06(−0.05 to 0.18)	
HOMA2-R	PEMA (*n* = 141)	2.11 (1.53, 2.95)	^a^ 2.01 (1.43, 2.77)	1.92 (1.45, 2.66) *	−0.20(−0.33 to −0.20)	0.017
	CTRL (*n* = 138)	1.99 (1.47, 2.82)	^b^ 1.98 (1.51, 2.74)	1.94 (1.47, 2.87)	0.03(−0.08 to 0.11)	
HOMA2-β	PEMA (*n* = 141)	75.0 (57.2, 95.3)	^a^ 69.0 (52.0, 84.8)	69.4 (55.5, 91.7)	−3.9(−5.6 to −1.0)	0.451
	CTRL (*n* = 138)	75.9 (59.1, 94.0)	^b^ 78.0 (55.9, 100.1)	71.3 (54.4, 93.0)	−1.1(−3.0 to 1.5)	
Disposition index	PEMA (*n* = 141)	0.36 (0.28, 0.46)	^a^ 0.35 (0.26, 0.47)	0.36 (0.27, 0.53)	0.02(−0.01 to 0.04)	0.030
	CTRL (*n* = 138)	0.37 (0.27, 0.48)	^b^ 0.36 (0.30, 0.47)	0.36 (0.27, 0.47)	−0.01(−0.04 to 0.01)	

Data are shown as mean ± SD, median (25% percentile, 75% percentile), or mean or median change (95% confidence interval). * *p* < 0.05 versus Week 0, paired *t*-test. ^a^ Data were obtained in 132 patients. ^b^ Data were obtained in 133 patients. PEMA, pemafibrate; CTRL, control; HbA1c, glycated hemoglobin; HOMA, homeostatic model assessment.

**Table 3 pharmaceutics-15-01838-t003:** Relationships between changes in indices for glucose metabolism associated with pemafibrate and changes in metabolic parameters in the 52-week study period.

	ΔHOMA-2R		ΔDI	
Variables	*ρ*	*p*-Value	*ρ*	*p*-Value
ΔBMI (kg/m^2^)	0.119	0.160	−0.018	0.037
ΔTriglyceride (mg/dL)	0.265	0.002	−0.258	0.002
ΔHDL-C (ng/mL)	−0.190	0.002	0.130	0.124
ΔAST (IU/L)	0.017	0.838	0.032	0.709
ΔALT (IU/L)	0.148	0.079	−0.142	0.093
Δγ-GTP (IU/L)	0.329	<0.001	−0.335	<0.001
ΔeGFR (mL/min/1.73 m^2^)	0.017	0.838	−0.161	0.058

*p*-values were obtained by Spearman’s rank correlation analysis. HOMA, homeostatic model assessment; DI, disposition index; BMI, body mass index; HDL-C, high-density lipoprotein cholesterol; AST, aspartate aminotransferase; ALT, alanine aminotransferase; γ-GTP, γ-glutamyl transpeptidase; eGFR, estimated glomerular filtration rate.

## Data Availability

The data that support the findings of this study are available from the corresponding author upon reasonable request.

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
