# Peer review of "Favorable Effect of Pemafibrate on Insulin Resistance and β-Cell Function in Subjects with Type 2 Diabetes and Hypertriglyceridemia: A Subanalysis of the PARM-T2D Study"

_pharmaceutics, 2023, doi:10.3390/pharmaceutics15071838_

Round 1
Reviewer 1 Report
Reviewed manuscript by Hiroshi Nomoto et al. is very important for diabetologists and patients with T2DM. There are only two suggestions for Authors:
1. Line 53. It is"... and sodium-glucose co-transporter 2 ....." I think, it will be better if this sequence will contain information "... and sodium-glucose co-transporter 2 (SGLT2)...." Name "SGLT2" is better known than its full name. Please include this "name"
2. There are differences in number of participants. Lines 116-123: 685 - 35 - 226 - 103 = 321 (Not 279). I think, that according to information presented in Figure 1, these values should be as below: ".. 35 participants who did not meet inclusion criteria and 268 patients...." If so, 685 - 35 - 268 - 103 = 279.
Additional comments to review of manuscript entitled “Favorable effect of pemafibrate on glucose metabolism in subjects with type 2 diabetes and hypertriglyceridemia: a subanalysis of the PARM-T2D study”
by Nomoto H, Kito K., Iesaka H, et al.
Insulin resistance, reduced insulin secretory capacity and dyslipidemia are pathologies observed in patients with type 2 diabetes mellitus (T2DM). The beneficial effect on metabolism of lipids shows pemafibrate, a selective modulator of peroxisome proliferator-activated receptors (PPAR-α). It is a potent and highly selected agonist for human PPAR-α, which plays a role in lipid metabolism. For example, a phase 3 clinical trial revealed that fibrates effectively decrease serum triglyceride (TG) and increase serum high-density lipoprotein cholesterol (HDL-C). This observation suggests, that fibrates, such as pemafibrate may be a therapeutic agent for patients with T2DM. Unfortunately, information on its effect on glucose metabolism remains scarce.
The aim of presented study was clarifying the effects of pemafibrate on glucose metabolism. This was analysis of Author’s previous multicenter prospective observational PARM-T2D study comparing the effect of pemafibrate with conventional therapies. As participants, were 685 patients with T2DM and hypertriglyceridemia. Based on excluded criteria, only 279 participants (141 in pemafibrate group and 138 as a control group) met the criteria for the subanalysis. Fasting blood and urine samples and physical assessments were evaluated at baseline and weeks 12, 24, and 52. The effect of investigated fibrate on glucose metabolism was observed based on fasting serum C-peptide, changes of HbA1c, HOMA2-R, and disposition index. These studies were conducted at nine specialized centers for treatment of diabetes located in Hokkaido (PARM-T2D study cohort).
The protocol was approved by the Institutional Review Board of Hokkaido University Hospital Clinic Research and Medical Innovation Center, and the study was performed according to the principles of the Declaration of Helsinki.
Results obtained from investigated and control groups were compared using adequate statistical analysis, such as a paired t-test, the Wilcoxon signed-rank test, analysis of covariance (ANOVA), multivariate analysis, etc. Obtained results are presented in text and 3 tables. Included tables cause that results are more easy and friendly for readers.
Discussion is performed based on obtained results in comparison with results obtained by other researchers. Based on obtained results, Authors suggest that PPAR-α modulator used in research, can ameliorate insulin resistance and retain
β-cell function in patients with T2DM and hypertriglyceridemia.
The conclusions are consistent with the evidence and presented arguments.
The cited references are appropriate and about of 40 per cent of them are published within the last 5 years.
Included 3 tables and 2 figures are appropriate and they are easy to interpret and understand.
Presented publication will be an important source of knowledge on this subject for oncologists, pharmacists, as well as researchers.
Overall Recommendation: “Accept after Minor Revision”. Suggestions of corrections are mentioned in “Information for Authors”.
Reviewer 2 Report
The authors explored the effect of pemafibrate on glucose metabolism and insulin resistance parameters in subjects with type 2 diabetes and hypertriglyceridemia. The study is designed as a secondary analysis of the PARM-T2D study. The topic of the manuscript is interesting, it brings a contribution to the field. The manuscript is well written, and the English language is fluent. However, some comments regarding the manuscript are written below.
-The title of the manuscript should be rewritten in order to better reflect the main message of the study-the results describing the effects of pemafibrate on insulin resistance parameters.
-Which antidiabetic treatment did the patients in this subanalysis receive? This should be described in detail in the revised manuscript-similar to the S1 Table. I suggest moving at least some information from Table S1 to the original manuscript. Some of the patients received SGLT-2 inhibitors or GLP-1 receptor agonists-they are known to possess beneficial effects on insulin secretion and insulin resistance parameters-did the authors take into account the possible confounding effects? Do the authors have the information how long the patients received (stable) antidiabetic treatment before the inclusion to the study?
-The authors should emphasize the time-dependency of the pemafibrate effect on HOMA-2R and DI.
-I suggest adding the perspectives for future research of pemafibrate and also possible clinical implications (metabolic syndrome, NASH, etc.) according to the results of the present study to the discussion section of the revised manuscript.
Reviewer 3 Report
My compliments with the authors for the research presented but introduction must be improved both regarding the type 2 diabetes and the risk of cardiovascular disease as presented in "resent pharmacological options in type 2 diabetes and synergic mechanism in cardiovascular disease" and how is a risk of death as in atherogenic dyslipidemia on admission is associated with poorer outcome in people with and without diabetes hospitalized for covid-19".
Do you have any data regarding the pharmacological treatment of the patients for type 2 diabetes? have you a sub analysis correlating the use of pemafibrate and the treatment of diabetes? (GLP1-RA, SGLT2i, insulin o metformin treatment?) Please also add a comment regarding the various treatments of type 2 diabetes and how some of them can increase the risk for hospitalization as in "severe hypoglycemia in patients with known diabetes requiring emergency department care: a report from an italian multicenter study"
What is the main question addressed by the research?
The authors evaluate the use of pemafibrate and its effects on glucose metabolismi in patients with type 2 diabetes. It’s innovative as research and the interesting for the potential effects is important as type 2 diabetes is considered a disease with important impact in peoples life. Prevent it will be important.
2. Do you consider the topic original or relevant in the field? Does it address a specific gap in the field?
Yes it’s original
3. What does it add to the subject area compared with other published material?
Presents the possibility of a new metabolic effect of pemifibrate in people with diabetes and opens a channel for prediabetes.
4. What specific improvements should the authors consider regarding the methodology? What further controls should be considered?
Do you have any data regarding the pharmacological treatment of the patients for type 2 diabetes? have you a sub analysis correlating the use of pemafibrate and the treatment of diabetes? (GLP1-RA, SGLT2i, insulin o metformin treatment?)
6. Are the references appropriate?
Yes but must be improved specially in the introduction as already reported in the first review, regarding the type 2 diabetes, definition and the risk of cardiovascular disease as presented in "resent pharmacological options in type 2 diabetes and synergic mechanism in cardiovascular disease" and how diabetes is a risk of death as in "atherogenic dyslipidemia on admission is associated with poorer outcome in people with and without diabetes hospitalized for covid-19”. We must not forget how during the pandemia of COVID 19 the incidence of mortality for the secondary complications is important in patients with type 2 diabetes. Please also add a comment regarding the various treatments of type 2 diabetes and how some of them can increase the risk for hospitalization as in "severe hypoglycemia in patients with known diabetes requiring emergency department care: a report from an italian multicenter study"
7. Please include any additional comments on the tables and figures.
Add a sub analysis regarding the pharmacological treatment of the patients with type 2 diabetes and the respective results.
minor editing regarding the english and some correction in grammatic
